# TEMPORAL DOMAIN GENERALIZATION WITH DRIFT-AWARE DYNAMIC NEURAL NETWORKS

**Guangji Bai,**[*] **Chen Ling**[*] **& Liang Zhao** [†]
Department of Computer Science
Emory University, Atlanta, GA, USA

## ABSTRACT

Temporal domain generalization is a promising yet extremely challenging area where the goal is to learn models under temporally changing data distributions and generalize to unseen data distributions following the trends of the change. The advancement of this area is challenged by: 1) characterizing data distribution drift and its impacts on models, 2) expressiveness in tracking the model dynamics, and 3) theoretical guarantee on the performance. To address them, we propose a Temporal Domain Generalization with Drift-Aware Dynamic Neural Network (DRAIN) framework. Specifically, we formulate the problem into a Bayesian framework that jointly models the relation between data and model dynamics. We then build a recurrent graph generation scenario to characterize the dynamic graph-structured neural networks learned across different time points. It captures the temporal drift of model parameters and data distributions and can predict models in the future without the presence of future data. In addition, we explore theoretical guarantees of the model performance under the challenging temporal DG setting and provide theoretical analysis, including uncertainty and generalization error. Finally, extensive experiments on several real-world benchmarks with temporal drift demonstrate the proposed method's effectiveness and efficiency.

## 1 INTRODUCTION

In machine learning, researchers often assume that training and test data follow the same distribution for the trained model to work on test data with some generalizability. However, in reality, this assumption usually cannot be satisfied, and when we cannot make sure the trained model is always applied in the same domain where it was trained. This motivates Domain Adaptation (DA) which builds the bridge between source and target domains by characterizing the transformation between the data from these domains (Ben-David et al., 2010; Ganin et al., 2016; Tzeng et al., 2017). However, in more challenging situations when target domain data is unavailable (e.g., no data from an unknown area, no data from the future, etc.), we need a more realistic scenario named Domain Generalization (DG) (Shankar et al., 2018; Arjovsky et al., 2019; Dou et al., 2019).

Most existing works in DG focus on generalization among domains with categorical indices, such as generalizing the trained model from one dataset (e.g., MNIST (LeCun et al., 1998)) to another (e.g., SVHN (Netzer et al., 2011)), from one task (e.g., image classification (Krizhevsky et al., 2012)) to another (e.g., image segmentation (Lin et al., 2014)), etc. However, in many real-world applications, the "boundary" among different domains is unavailable and difficult to detect, leading to a *concept drift* across the domains. For example, when a bank leverages a model to predict whether a person will be a "defaulted borrower", features like "annual incoming", "profession type", and "marital status" are considered. However, due to the temporal change of the society, how these feature values indicate the prediction output should change accordingly following some trends that could be predicted somehow in a range of time. Figure 1 shows another example, seasonal flu prediction via Twitter data which evolves each year in many aspects. For example, monthly active users are increasing, new friendships are formed, the age distribution is shifting under some trends, etc. Such temporal change in data distribution gradually outdated the models. Correspondingly, suppose there was an ideal, always update-to-date model, then the model parameters should gradually change

---

[*]Equal contribution. {guangji.bai, chen.ling}@emory.edu
[†]Corresponding author. liang.zhao@emory.edu

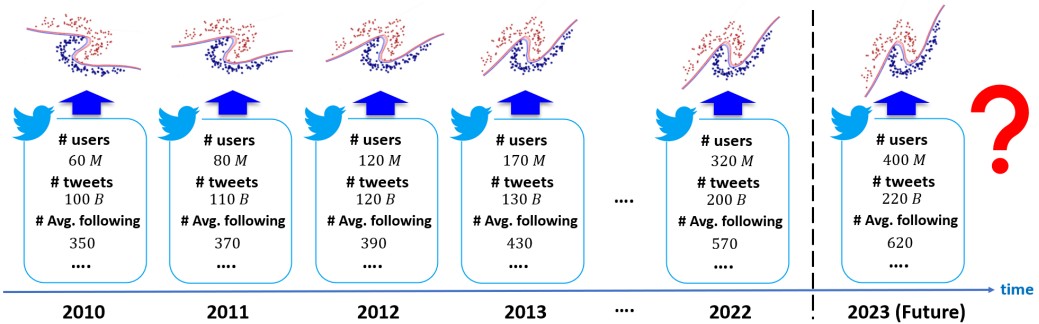

Figure 1: **An illustrative example of temporal domain generalization.** Consider training a model for some classification tasks based on the annual Twitter dataset such that the trained model can generalize to the future domains (e.g., 2023). The temporal drift of data distribution can influence the prediction model such as the rotation of the decision boundary in this case.

correspondingly to counter the trend of data distribution shifting across time. It can also "predict" what the model parameters should look like in an arbitrary (not too far) future time point. This requires the power of *temporal* domain generalization.

However, as an extension of traditional DG, temporal DG is extremely challenging yet promising. Existing DG methods that treat the domain indices as a categorical variable may not be suitable for temporal DG as they require the domain boundary as apriori to learn the mapping from source to target domains (Muandet et al., 2013; Motiian et al., 2017; Balaji et al., 2018; Arjovsky et al., 2019). Until now, temporal domain indices have been well explored only in DA (Hoffman et al., 2014; Ortiz-Jimenez et al., 2019; Wang et al., 2020) but not DG. There are very few existing works in temporal DG due to its big challenges. One relevant work is *Sequential Learning Domain Generalization* (S-MLDG) (Li et al., 2020) that proposed a DG framework over sequential domains via meta-learning (Finn et al., 2017). S-MLDG meta-trains the target model on all possible permutations of source domains, with one source domain left for meta-test. However, S-MLDG in fact still treats domain index as a categorical variable, and the method was only tested on categorical DG dataset. A more recent paper called *Gradient Interpolation* (GI) (Nasery et al., 2021) proposes a temporal DG algorithm to encourage a model to learn functions that can extrapolate to the near future by supervising the first-order Taylor expansion of the learned function. However, GI has very limited power in characterizing model dynamics because it can only learn how the activation function changes along time while making all the remaining parameters fixed across time.

The advancement of temporal domain generalization is challenged by several critical bottlenecks, including **1) Difficulty in characterizing the data distribution drift and its influences on models.** Modeling the temporally evolving distributions requires making the model time-sensitive. Intuitive ways include feeding the time as an input feature to the model, which is well deemed simple yet problematic as it discards the other features' dependency on time and dependency on other *confounding* factors changed along time (Wang et al., 2020). Another possible way is to make the model parameters a function of time. However, these ways cannot generalize the model to future data as long as the whole model's dynamics and data dynamics are not holistically modeled. **2) Lack of expressiveness in tracking the model dynamics.** Nowadays, complex tasks have witnessed the success of big complex models (e.g., large CNNs (Dosovitskiy et al., 2020)), where the neurons and model parameters are connected as a complex graph structure. However, they also significantly challenge tracking their model dynamics in temporal DG. An expressive model dynamics characterization and prediction requires mapping data dynamics to *model dynamics* and hence the graph dynamics of model parameters across time. This is a highly open problem, especially for the temporal DG area. **3) Difficulty in theoretical guarantee on the performance.** While there are fruitful theoretical analyses on machine learning problems under the independent and identically distributed assumptions (He & Tao, 2020), similar analyses meet substantial hurdles to be extended to out-of-distribution (*OOD*) problem due to the distribution drift over temporally evolving domains. Therefore, it is essential to enhance the theoretical analyses on the model capacity and theoretical relation among different temporal domain generalization models.

To address all the above challenges, we propose a Temporal Domain Generalization with **DR**ift-**A**ware dynam**I**c neural **N**etworks (**DRAIN**) framework that solves all challenges above simultane-

ously. Specifically, we propose a generic framework to formulate temporal domain generalization by a Bayesian treatment that jointly models the relation between data and model dynamics. To instantiate the Bayesian framework, a recurrent graph generation scenario is established to encode and decode the dynamic graph-structured neural networks learned across different timestamps. Such a scenario can achieve a fully time-sensitive model and can be trained in an end-to-end manner. It captures the temporal drift of model parameters and data distributions, and can predict the models in the future *without* the presence of future data.

**Our contributions include: 1)** We develop a novel and adaptive temporal domain generalization framework that can be trained in an end-to-end manner. **2)** We innovatively treat the model as a dynamic graph and leverage graph generation techniques to achieve a fully time-sensitive model. **3)** We propose to use the sequential model to learn the temporal drift adaptively and leverage the learned sequential pattern to predict the model status on the future domain. **4)** We provide theoretical analysis on both uncertainty quantification and generalization error of the proposed method. **5)** We demonstrate our model's efficacy and superiority with extensive experiments.

## 2 RELATED WORK

**Continuous Domain Adaptation.** Domain Adaptation (DA) has received great attention from researchers in the past decade (Ben-David et al., 2010; Ganin et al., 2016; Tzeng et al., 2017) and readers may refer to (Wang & Deng, 2018) for a comprehensive survey. Under the big umbrella of DA, continuous domain adaptation considers the problem of adapting to target domains where the domain index is a continuous variable (temporal DA is a special case when the domain index is 1D). Approaches to tackling such problems can be broadly classified into three categories: (1) biasing the training loss towards future data via transportation of past data (Hoffman et al., 2014; Ortiz-Jimenez et al., 2019), (2) using time-sensitive network parameters and explicitly controlling their evolution along time (Kumagai & Iwata, 2016; 2017; Mancini et al., 2019), (3) learning representations that are time-invariant using adversarial methods (Wang et al., 2020). The first category augments the training data, the second category reparameterizes the model, and the third category redesigns the training objective. However, data may not be available for the target domain, or it may not be possible to adapt the base model, thus requiring Domain Generalization.

**Domain Generalization (DG).** A diversity of DG methods have been proposed in recent years (Muandet et al., 2013; Motiian et al., 2017; Li et al., 2017a; Balaji et al., 2018; Dou et al., 2019; Nasery et al., 2021; Yu et al., 2022). According to (Wang et al., 2021), existing DG methods can be categorized into the following three groups, namely: (1) *Data manipulation:* This category of methods focuses on manipulating the inputs to assist in learning general representations. There are two kinds of popular techniques along this line: a). Data augmentation (Tobin et al., 2017; Tremblay et al., 2018), which is mainly based on augmentation, randomization, and transformation of input data; b). Data generation (Liu et al., 2018; Qiao et al., 2020), which generates diverse samples to help generalization. (2) *Representation learning:* This category of methods is the most popular in domain generalization. There are two representative techniques: a). Domain-invariant representation learning (Ganin et al., 2016; Gong et al., 2019), which performs kernel, adversarial training, explicitly features alignment between domains, or invariant risk minimization to learn domain-invariant representations; b). Feature disentanglement (Li et al., 2017b), which tries to disentangle the features into domain-shared or domain-specific parts for better generalization. (3) *Learning strategy:* This category of methods focuses on exploiting the general learning strategy to promote the generalization capability, e.g, ensemble learning (Mancini et al., 2018), meta-learning (Dou et al., 2019), gradient operation (Huang et al., 2020), etc.

Existing works above consider generalization across categorical domains, while in this paper, we assume the domain index set is across time (namely, temporal), and the domain shifts smoothly over time. Unfortunately, there is only very little work under this setting. The first work called *Sequential Learning Domain Generalization* (S-MLDG) (Li et al., 2020) proposed a DG framework over sequential domains based on the idea of meta-learning. A more recent work called *Gradient Interpolation* (GI) (Nasery et al., 2021) proposes a temporal DG algorithm to encourage a model to learn functions that can extrapolate well to the near future by supervising the first-order Taylor expansion of the learned function. However, neither work can adaptively learn the temporal drift across the domains while keeping the strong expressiveness of the learned model.

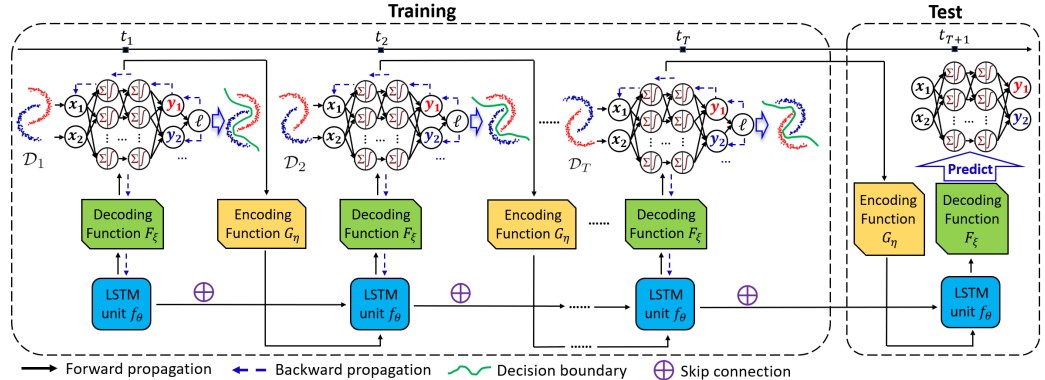

Figure 2: A high-level overview of our DRAIN framework. Best viewed in color.

## 3 METHODOLOGY

In this section, we first provide the problem formulation of temporal domain generalization and then introduce our proposed framework, followed by our theoretical analyses.

### 3.1 PROBLEM FORMULATION

**Temporal Domain Generalization.** We consider prediction tasks where the data distribution evolves with time. During training, we are given $T$ observed source domains $\mathcal{D}_1, \mathcal{D}_2, \cdots, \mathcal{D}_T$ sampled from distributions on $T$ arbitrary time points $t_1 \leq t_2 \leq \cdots \leq t_T$, with each $\mathcal{D}_s = \{(\mathbf{x}_i^{(s)}, y_i^{(s)}) \in \mathcal{X}_s \times \mathcal{Y}_s\}_{i=1}^{N_s}$, $s = 1, 2, \cdots, T$ where $\mathbf{x}_i^{(s)}, y_i^{(s)}$ and $N_s$ denotes the input feature, label and sample size at timestamp $t_s$, respectively, and $\mathcal{X}_s, \mathcal{Y}_s$ denotes the input feature space and label space at timestamp $t_s$, respectively. The trained model will only be tested on some target domain *in the future*, i.e., $\mathcal{D}_{T+1}$ where $t_{T+1} \geq t_T$. Our setting further assumes the existence of concept drift across different domains, i.e., the domain distribution is changing across time by following some patterns.

Our *goal* is to build a model that proactively captures the concept drift. Given labeled data from the source domains $\mathcal{D}_1, \mathcal{D}_2, \cdots, \mathcal{D}_T$, we learn the mapping function $g_{\boldsymbol{\omega}_s} : \mathcal{X}_s \rightarrow \mathcal{Y}_s$ on each domain $\mathcal{D}_s$, $s = 1, 2, \cdots, T$ where $\boldsymbol{\omega}_s$ denotes the function parameters at timestamp $t_s$, respectively, and then predict the dynamics across the parameters $\boldsymbol{\omega}_1, \boldsymbol{\omega}_2, \cdots, \boldsymbol{\omega}_T$. Finally, we predict the parameters $\boldsymbol{\omega}_{T+1}$ for the mapping function $g_{\boldsymbol{\omega}_{T+1}} : \mathcal{X}_{T+1} \rightarrow \mathcal{Y}_{T+1}$ on the unseen future domain. As shown in Figure 1, due to the temporal drift in data distribution, e.g. the input features such as Twitter user age distribution and number of tweets increase each year, the prediction model is expected to evolve accordingly, e.g. the magnitude of model parameter weights will decrease annually. Despite the necessity, handling the above problem is an open research area due to several existing challenges: 1) Difficulty in characterizing data distribution drift as well as how it influences the model. 2) Lack of expressiveness in automatically capturing the dynamics of how neural network evolves across time. 3) Theoretical guarantee on model's performance (e.g., generalization error, uncertainty) on future domains is hard to obtain due to the unknown and (potentially) complicated concept drift.

### 3.2 PROPOSED METHOD

In this section, we introduce how we address the challenges mentioned above. For the first challenge, we build a systematic Bayesian probability framework to represent the concept drift over the domains, which instantly differentiates our work from all existing methods in DG. For the second challenge, we propose modeling a neural network with changing parameters as a dynamic graph and achieving a temporal DG framework that can be trained end-to-end by graph generation techniques. We further improve the proposed method's generalization ability by introducing a skip connection module over different domains. Finally, to handle the last challenge, we explore theoretical guarantees of model performance under the challenging temporal DG setting and provide theoretical analyses of our proposed method, such as uncertainty quantification and generalization error.

#### 3.2.1 A PROBABILISTIC VIEW OF CONCEPT DRIFT IN TEMPORAL DOMAIN GENERALIZATION

To perform domain generalization over temporally indexed domains, we need to capture the concept drift within a given time interval. From a probabilistic point of view, for each domain $\mathcal{D}_s$, $s =$

$1, 2, \cdots, T$, we can learn a neural network $g_{\boldsymbol{\omega}_s}$ by maximizing the conditional probability $\Pr(\boldsymbol{\omega}_s|\mathcal{D}_s)$, where $\boldsymbol{\omega}_s$ denotes the status of model parameters at timestamp $t_s$. Due to the evolving distribution of $\mathcal{D}_s$, the conditional probability $\Pr(\boldsymbol{\omega}_s|\mathcal{D}_s)$ will change over time accordingly. Our ultimate goal is to predict $\boldsymbol{\omega}_{T+1}$ given all training data $\mathcal{D}_1, \mathcal{D}_2, \cdots, \mathcal{D}_T$ ($\mathcal{D}_{1:T}$ for short), i.e., $\Pr(\boldsymbol{\omega}_{T+1}|\mathcal{D}_{1:T})$. By the Law of total probability, we have

$$\Pr\left(\boldsymbol{\omega}_{T+1} \mid \mathcal{D}_{1:T}\right) = \int_{\Omega} \underbrace{\Pr\left(\boldsymbol{\omega}_{T+1} \mid \boldsymbol{\omega}_{1:T}, \mathcal{D}_{1:T}\right)}_{\text{inference}} \cdot \underbrace{\Pr\left(\boldsymbol{\omega}_{1:T} \mid \mathcal{D}_{1:T}\right)}_{\text{training}} d\boldsymbol{\omega}_{1:T}, \tag{1}$$

where $\Omega$ is the space for $\boldsymbol{\omega}_{1:T}$. The first term in the integral represents the inference phase, i.e., how we predict the status of the target neural network in the future (namely, $\boldsymbol{\omega}_{T+1}$) given all history statuses, while the second term denotes the training phase, i.e., how we leverage all source domains' training data $\mathcal{D}_{1:T}$ to obtain the status of the neural network on each source domain, namely $\boldsymbol{\omega}_{1:T}$. By the chain rule of probability, we can further decompose the training phase as follows:

$$\begin{aligned} \Pr\left(\boldsymbol{\omega}_{1:T} \mid \mathcal{D}_{1:T}\right) &= \prod_{s=1}^{T} \Pr\left(\boldsymbol{\omega}_s \mid \boldsymbol{\omega}_{1:s-1}, \mathcal{D}_{1:T}\right) \\ &= \Pr\left(\boldsymbol{\omega}_1 \mid \mathcal{D}_1\right) \cdot \Pr\left(\boldsymbol{\omega}_2 \mid \boldsymbol{\omega}_1, \mathcal{D}_{1:2}\right) \cdots \Pr\left(\boldsymbol{\omega}_T \mid \boldsymbol{\omega}_{1:T-1}, \mathcal{D}_{1:T}\right). \end{aligned} \tag{2}$$

Here we assume for each time point $t_s$, the model parameter $\boldsymbol{\omega}_s$ only depends on the current and previous domains (namely, $\{\mathcal{D}_i : i \leq s\}$), and there is no access to future data (even unlabeled). Now we can break down the whole training process into $T-1$ steps, where each step corresponds to learning the model parameter on the new domain conditional on parameter statuses from the history domains and training data, i.e., $\Pr\left(\boldsymbol{\omega}_{s+1} \mid \boldsymbol{\omega}_{1:s}, \mathcal{D}_{1:s}, \mathcal{D}_{s+1}\right)$, $\forall s < T$.

### 3.2.2 NEURAL NETWORK WITH DYNAMIC PARAMETERS

Since the data distributions change temporally, the parameter $\boldsymbol{\omega}_s$ in $g_{\boldsymbol{\omega}_s}$ needs to be updated accordingly to address the temporal drift across the domains. In this work, we consider leveraging *dynamic graphs* to model the temporally evolving neural networks in order to retain maximal expressiveness.

Intuitively, a neural network $g_{\boldsymbol{\omega}}$ can be represented as an edge-weighted graph $G = (V, E, \psi)$, where each node $v \in V$ represents a neuron of $g_{\boldsymbol{\omega}}$ while each edge $e \in E$ corresponds to a connection between two neurons in $g_{\boldsymbol{\omega}}$. Moreover, given a connection $e$ between neuron $u$ and $v$, i.e., $e = (u, v) \in E$, function $\psi : E \to \mathbb{R}$ denotes the weight parameter between these two neurons, i.e., $\psi(u, v) = w_{u,v}, \forall (u, v) \in E$. Essentially, $\boldsymbol{\omega} = \psi(E) = \{w_{u,v} : (u, v) \in E\}$ is a set of parameter values indexed by all edges in E and $\boldsymbol{\omega}$ represents the entire set of parameters for neural network $g$. Notice that we give a general definition of $g_{\boldsymbol{\omega}}$ so that both shallow models (namely, linear model) and deep neural networks (e.g., MLP, CNN, RNN, GNN) can be treated as special cases here. We aim to characterize the potential drift across domains by optimizing and updating the graph structure (i.e., edge weight) of $g_{\boldsymbol{\omega}}$. You et al. (2020) have proven that optimizing the graph structure of the neural network could have a smaller search space and a more smooth optimization procedure than exhaustively searching over all possible connectivity patterns.

We consider the case where the architecture or topology of neural network $g_{\boldsymbol{\omega}}$ is given, i.e., $V$ and $E$ are fixed, while the parameter $\boldsymbol{\omega}$ is changing constantly w.r.t time point $t_s$. In this sense, we can write $\boldsymbol{\omega}_s = \psi(E|s)$ where $\psi(\cdot|s)$ (*abbrev.* $\psi_s$) depends only on time point $t_s$. Now the triplet $G = (V, E, \psi_s)$ defines a *dynamic graph* with evolving edge weights.

### 3.2.3 END-TO-END LEARNING OF CONCEPT DRIFT

Given history statuses $\{\boldsymbol{\omega}_{1:s}\}$ of the neural network learned from $\{\mathcal{D}_{1:s}\}$, we aim at generalizing and extrapolating $\boldsymbol{\omega}_{s+1}$ so that it produces good performance on the new domain $\mathcal{D}_{s+1}$ in an end-to-end manner. In fact, by viewing the neural networks $\{\boldsymbol{\omega}_{1:s}\}$ as dynamically evolving graphs, a natural choice is to characterize the latent graph distribution of $\{\boldsymbol{\omega}_{1:s}\}$ by learning from its evolving trend. Consequently, $\boldsymbol{\omega}$'s can be directly sampled from the distribution for the prediction in future domains.

We characterize the latent distribution of $\{\boldsymbol{\omega}_{1:s}\}$ as a sequential learning process based on a recurrent architecture, and each unit $f_{\theta}$ in the recurrent model is parameterized by $\theta$ to generate $\boldsymbol{\omega}_s$ by accounting for previous $\{\boldsymbol{\omega}_i : i < s\}$. Specifically, at each recurrent block (i.e., time step) $t_s$, $f_{\theta}$ produces two outputs $(m_s, h_s)$, where $m_s$ is the current memory state and $h_s$ is a latent probabilistic distribution (i.e., hidden output of $f_{\theta}$) denoting the information carried from previous time steps. The latent probabilistic distribution $h_t$ allows us to generate the dynamic graph $\boldsymbol{\omega}_s$ by a decoding function $F_{\xi}(\cdot)$. Intuitively, different from existing works that train and regularize a neural

network on single domain (Nasery et al., 2021), here we focus on directly searching for distribution of networks with "good architectures". Lastly, the sampled $\omega_s$ is encoded by a graph encoding function $G_\eta(\cdot)$, which then serves as the input of next recurrent block. Such a recurrent model is trained on a single domain $\mathcal{D}_s$ to generate $\omega_s$ for prediction by minimizing the empirical loss, i.e., $\min_{\theta,\xi,\eta} \sum_{i=1}^{N_s} \ell\big(g_{\omega_s}(\mathbf{x}_i^{(s)}), y_i^{(s)}\big)$, where $\ell(\cdot, \cdot)$ can be cross-entropy for classification or MSE for regression. The optimal $\omega_s$ on domain $\mathcal{D}_s$ will then be fed into the next domain $\mathcal{D}_{s+1}$ along with the memory state $m_s$ as input to guide the generation of $\omega_{s+1}$ until the entire training phase is done. For the inference phase, we feed the optimal parameters from the last training domain, namely $\omega_T$, into the encoding function and leverage the recurrent block, together with the memory state $m_T$ to predict the latent vector on the future domain $\mathcal{D}_{T+1}$, followed by the decoding function to decode the latent vector and generate the optimal parameters $\omega_{T+1}$.

### 3.2.4 LESS FORGETTING AND BETTER GENERALIZATION

During the training of recurrent models, it is also likely to encounter the performance degradation problem. Such a problem can be severe in temporal DG since a more complicated concept correlation exists between each domain. In addition, if the training procedure on each domain $\mathcal{D}_s$ takes a large number of iterations to converge, we may also observe the forgetting phenomenon (i.e., the recurrent model $f_\theta$ will gradually focus on the current training domain and have less generalization capability for future domains). To alleviate such a phenomenon, we leverage a straightforward technique - skip connection to bridge the training on $\mathcal{D}_s$ with previous domains $\{\mathcal{D}_{1:s-1}\}$. Specifically,

$$\Phi\Big(\omega_s, \big\{\omega_{s-\tau:s-1}\big\}\Big) \coloneqq \omega_s + \lambda \cdot \sum\nolimits_{i=s-\tau}^{s-1} \omega_i, \tag{3}$$

where $\lambda$ is regularization coefficient and $\tau$ denotes the size of the *sliding window*. The skip connection could enforce the generated network parameters $\omega_s$ to contain part of previous network's information, and the implementation of the fixed-sized sliding window can better alleviate the potential drawback of the computational cost. We summarize the overall generative process in Appendix A.2.

### 3.3 THEORETICAL ANALYSES

In this section, we provide a theoretical analysis of our proposed framework's performance in the target domain. Our analyses include uncertainty quantification and generalization error. Uncertainty characterizes the dispersion or error of an estimate due to the noise in measurements and the finite size of data sets, and smaller uncertainty means less margin of error over the model predictions. On the other hand, generalization error measures how accurate the model's prediction is on unseen data. Our analyses show that our proposed DRAIN achieves **both better prediction accuracy as well as smaller margin of error on target domain** compared with online and offline DG baselines.

First, we introduce two DG methods, namely online baseline and offline baseline as defined below:

**Definition 1.** Given timestamp $t_{s+1}$ and domains $\mathcal{D}_1, \mathcal{D}_2, \cdots, \mathcal{D}_{s+1}$, and model parameter state from previous timestamp, namely $\omega_s$. Define online model $\mathcal{M}_{\text{on}}$ and offline model $\mathcal{M}_{\text{off}}$ as $\omega_{s+1} = \text{argmax}_{\omega_{s+1}} \Pr(\omega_{s+1}|\omega_s, \mathcal{D}_{s+1})$ and $\omega_{s+1} = \text{argmax}_{\omega_{s+1}} \Pr(\omega_{s+1}|\mathcal{D}_{1:s+1})$, respectively.

Offline method $\mathcal{M}_{\text{off}}$ is trained using ERM over all source domains, while online method $\mathcal{M}_{\text{on}}$ considers one-step finetuning over the model parameter on each new domain's dataset. Both $\mathcal{M}_{\text{off}}$ and $\mathcal{M}_{\text{on}}$ are time-oblivious, i.e., unaware of the concept drift over time.

**Assumption 1.** *Consider a parameterized function $q_\theta(\cdot)$ to approximate $P(\omega_{t+1}|\omega_t)$, which is the unknown ground-truth concept drift of the model parameter distribution. It is assumed that the prior over $q_\theta$ follows a normal distribution with: $\mathbb{E}[q_{\theta_0}(\omega)] = \omega$, $Var(q_{\theta_0}(\omega)) = \sigma_{\theta_0}^2$, $\forall\, \omega \in \Omega$.*

**Definition 2** (Predictive Distribution). Given training sample $\mathcal{D}_1, \mathcal{D}_2, \cdots, \mathcal{D}_T$, and input feature from future domain, namely $\mathbf{x}_{T+1}$, the predictive distribution can be defined as

$$\Pr\big(\hat{y} \mid \mathbf{x}_{T+1}, \mathcal{D}_{1:T}\big) = \int \Pr\big(\hat{y} \mid \mathbf{x}_{T+1}, \omega_{T+1}\big) \Pr\big(\omega_{T+1} \mid \mathcal{D}_{1:T}\big) d\omega_{T+1}. \tag{4}$$

Our first theorem below shows that by capturing the concept drift over the sequential domains, our proposed method always achieves the smallest uncertainty in prediction on the future domain.

**Theorem 1** (Uncertainty Quantification). *Given training domains $\mathcal{D}_1, \mathcal{D}_2, \cdots, \mathcal{D}_T$ where $Var(\mathcal{D}_i)$ is the same, we have the following inequality over each method's predictive uncertainty, i.e., the variance of predictive distribution as defined in Eq. 4: $Var(\mathcal{M}_{ours}) < Var(\mathcal{M}_{on}) \leq Var(\mathcal{M}_{off})$.*

Table 1: Performance comparison of all methods in terms of misclassification error (in %) for classification tasks and mean absolute error (MAE) for regression tasks (both smaller the better.) Results of comparison methods on all datasets except "Appliance" are reported from Nasery et al. (2021). "-" denotes that the method could not converge on the specific dataset.

| Model | Classification (in %) | | | | | Regression | |
|---|---|---|---|---|---|---|---|
| | 2-Moons | Rot-MNIST | ONP | Shuttle | Elec2 | House | Appliance |
| Offline | $22.4 \pm 4.6$ | $18.6 \pm 4.0$ | $\mathbf{33.8 \pm 0.6}$ | $0.77 \pm 0.1$ | $23.0 \pm 3.1$ | $11.0 \pm 0.36$ | $10.2 \pm 1.1$ |
| LastDomain | $14.9 \pm 0.9$ | $17.2 \pm 3.1$ | $36.0 \pm 0.2$ | $0.91 \pm 0.18$ | $25.8 \pm 0.6$ | $10.3 \pm 0.16$ | $9.1 \pm 0.7$ |
| IncFinetune | $16.7 \pm 3.4$ | $10.1 \pm 0.8$ | $34.0 \pm 0.3$ | $0.83 \pm 0.07$ | $27.3 \pm 4.2$ | $9.7 \pm 0.01$ | $8.9 \pm 0.5$ |
| CDOT | $9.3 \pm 1.0$ | $14.2 \pm 1.0$ | $34.1 \pm 0.0$ | $0.94 \pm 0.17$ | $17.8 \pm 0.6$ | - | - |
| CIDA | $10.8 \pm 1.6$ | $9.3 \pm 0.7$ | $34.7 \pm 0.6$ | - | $14.1 \pm 0.2$ | $9.7 \pm 0.06$ | $8.7 \pm 0.2$ |
| GI | $3.5 \pm 1.4$ | $7.7 \pm 1.3$ | $36.4 \pm 0.8$ | $0.29 \pm 0.05$ | $16.9 \pm 0.7$ | $9.6 \pm 0.02$ | $8.2 \pm 0.6$ |
| **DRAIN** | $\mathbf{3.2 \pm 1.2}$ | $\mathbf{7.5 \pm 1.1}$ | $38.3 \pm 1.2$ | $\mathbf{0.26 \pm 0.05}$ | $\mathbf{12.7 \pm 0.8}$ | $\mathbf{9.3 \pm 0.14}$ | $\mathbf{6.4 \pm 0.4}$ |

Our second theorem shows that, besides uncertainty, our proposed method can also achieves smallest generalization error thanks to learning the concept drift.

**Definition 3.** Given predictive distribution in Eq. 4, as well as ground-truth label $y_{T+1}$ from the future domain, define the predictive or generalization error as $err := \ell(\mathbb{E}[P(\hat{y}|\mathbf{x}_{T+1}, \mathcal{D}_{1:T})], y_{T+1})$.

**Theorem 2** (Generalization Error). *Assume $g_\omega(\cdot)$ has Lipschitz constant with upper bound $L_{upper}$ and lower bound $L_{lower}$ w.r.t $\omega$. We have the following inequality over each method's predictive error defined above: $err(\mathcal{M}_{ours}) < err(\mathcal{M}_{on}) < err(\mathcal{M}_{off})$.*

**Complexity Analyses.** In our implementation, the encoding and decoding functions are instantiated as MLPs. The total number of parameters of the encoding and decoding functions is $\mathcal{O}(Nd + C)$, which is *linear* in $N$. Here $N$ is the number of parameters in predictive models (namely $\omega$), $d$ is the width (i.e., number of neurons) of the last hidden layer of the encoding and decoding functions, and $C$ denotes the number of parameters for all the layers before the last for the encoding and decoding functions. Additionally, in many situations, the first few layers of representation learning could be shared. Hence, we do not need to generate all the parameters in $\omega$, but just the last few layers.

## 4 EXPERIMENT

In this section, we present the performance of DRAIN against other state-of-the-art approaches with both quantitative and qualitative analysis. The experiments in this paper were performed on a 64-bit machine with 4-core Intel Xeon W-2123 @ 3.60GHz, 32GB memory and NVIDIA Quadro RTX 5000. Additional experiment settings and results (e.g., hyperparameter setting and scalability analysis) are demonstrated in the appendix.[1]

### 4.1 EXPERIMENT SETTING

**Datasets.** We compare with the following classification datasets: Rotated Moons (2-Moons), Rotated MNIST (Rot-MNIST), Online News Popularity (ONP), Electrical Demand (Elec2), and Shuttle; and the following regression datasets: House prices dataset (House), Appliances energy prediction dataset (Appliance). The first two datasets are synthetic, where the rotation angle is used as a proxy for time. The remaining datasets are real-world datasets with temporally evolving characteristics. Dataset details can be found at Appendix A.1.1.

**Comparison Methods.** We adopt three sets of comparison methods: **practical baselines** that do not consider the concept drift, including *1). Offline* that treats all source domains as a single domain, *2). LastDomain* that only employs the last training domain, and *3). IncFinetune* that sequentially trains on each training domain. **Continuous domain adaptation methods** that focus only on DA, including *1). CDOT* (Ortiz-Jimenez et al., 2019) that transports most recent labeled examples to the future, and *2). CIDA* (Wang et al., 2020) that specifically tackles the continuous DA problem; and one **temporal domain generalization method**: *GI* (Nasery et al., 2021).

All experiments are repeated 10 times for each method, and we report the average results and the standard deviation in the following quantitative analysis. More detailed description of each comparison method and the parameter setting can be found in Appendix A.1.2 and A.1.3, respectively.

---

[1]Our open-source code is available at `https://github.com/BaiTheBest/DRAIN`.

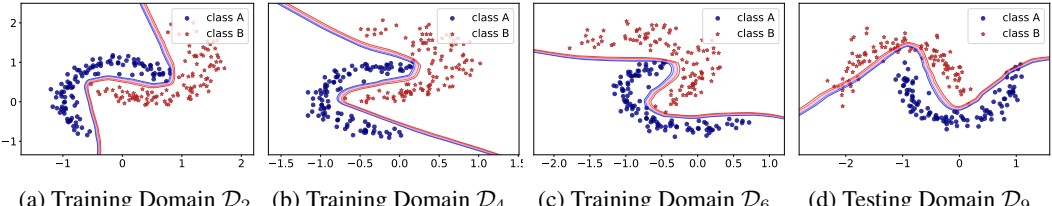

(a) Training Domain $\mathcal{D}_2$    (b) Training Domain $\mathcal{D}_4$    (c) Training Domain $\mathcal{D}_6$    (d) Testing Domain $\mathcal{D}_9$

Figure 3: **Visualization of the decision boundary of DRAIN** (blue dots and red stars represent different data classes). As the distribution of data points is consistently changing, as shown in Figure 3a - 3c, DRAIN can effectively characterize such a temporal drift and predict accurate decision boundaries on the unseen testing domain in Figure 3d.

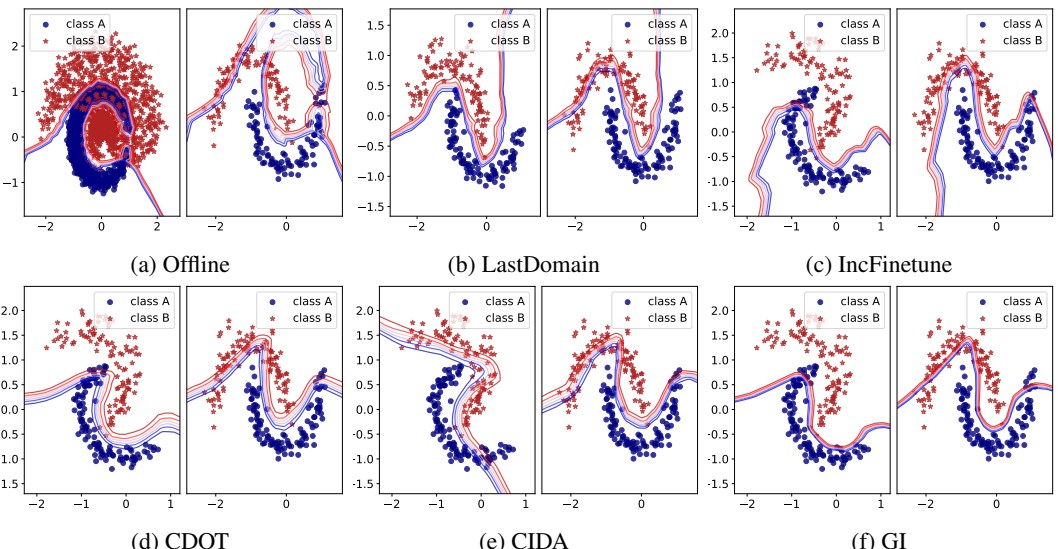

(a) Offline      (b) LastDomain      (c) IncFinetune

(d) CDOT        (e) CIDA        (f) GI

Figure 4: **Visualization of decision boundary** (blue dots and red stars represent different data classes), where the right subfigure of comparison methods Figure 4a - 4f demonstrate the decision boundary predicted for the test domain $\mathcal{D}_{T+1}$, the left subfigure in Figure 4a shows the decision boundary learned from the all data points in the concatenated training domain ($[\mathcal{D}_1, \cdots, \mathcal{D}_T]$), the left subfigure in Figure 4b shows the decision boundary learned from all samples in the last training domain $\mathcal{D}_T$, and the left subfigures in Figure 4c - 4f show the decision boundary learned on $\mathcal{D}_4$.

## 4.2 QUANTITATIVE ANALYSIS

We first illustrate the performance of our proposed method against comparison methods. The experiments are conducted in both classification and regression tasks with the domain generalization setting, i.e., models are trained on the training domains and deployed on the unseen testing domain.

As can be seen from Table 1, DRAIN consistently achieves competitive results across most datasets. Specifically, DRAIN excels the second-best approaches on Elec2 (CIDA), House (GI) and Appliance (GI) by a great margin. The only exception is the ONP dataset, where the Offline method achieves the best result and all state-of-the-art methods cannot generalize well on unseen testing domains since the ONP dataset does not exhibit a strong concept drift. Additionally, all time-oblivious baselines perform rather unsatisfactorily since they are not capable of handling the concept drift of the data distribution. Both CDOT and CIDA can generate better results than time-oblivious baselines, yet their generalization ability on the unseen domains is still limited as the maintained time-invariant representation in both methods cannot address the concept drift without any data in the testing domain. As the only method that addresses the temporal domain generalization problem, GI imposes a gradient regularization with a non-parametric activation function to handle the concept drift, which relies too much on the task-specific heuristic. In contrast, DRAIN proposes to sequentially model each domain in an end-to-end manner, which could address the concept drift more inherently.

### 4.3 QUALITATIVE ANALYSIS

We compare different methods qualitatively by visualizing the decision boundary on the 2-Moons dataset. As shown in Figure 3a - 3c, we demonstrate the decision boundary predicted by DRAIN at $\mathcal{D}_2$, $\mathcal{D}_4$, $\mathcal{D}_6$ training domains, and the final predicted decision boundary on the testing domain $\mathcal{D}_9$ (Figure 3d). As can be seen, DRAIN can successfully characterize the concept drift by sequentially modeling the $\{\mathcal{D}_T\}$, and the learned decision boundary could rotate correctly along time.

We further visualize the decision boundary learned by other comparison methods in Figure 4a - 4f. Firstly, the left subfigure in Figure 4a shows the decision boundary learned by the Offline method on the concatenated training domains $\{\mathcal{D}_{1:T}\}$, and the learned decision boundary overfits the training data and shows poor performance when generalizing on the unseen testing domain (the right subfigure of 4a). Furthermore, as the current state-of-the-art continuous domain adaptation methods, CDOT transports the most recent labeled data points in $\mathcal{D}_T$ to the future, which makes the learned decision boundary almost temporal-invariant (Figure 4d) and cannot generalize well in the scenario of domain generalization. CIDA utilizes the adversarial training technique to solve the domain adaptation, yet the predicted decision boundary in Figure 4e is less stable than other state-of-the-art methods due to its model complexity. Lastly, even though GI is the only method proposed to tackle the temporal domain generalization problem, the produced decision boundaries, as shown in both the training domain and testing domain (Figure 4f), are still less accurate than our proposed method, since they heavily utilize heuristics to regularize the gradient.

### 4.4 SENSITIVITY ANALYSIS

We conduct sensitivity analysis on the depth of the neural network $g_{\omega_s}$ for DRAIN. As shown in Figure 5, the optimal number of hidden layers for $g_{\omega_s}$ is 2 and 1 on 2-Moons and Electric dataset, respectively. The curve on both datasets has an inverse "U" shape, meaning that too few layers may limit the general expressiveness of our model, while too many layers could potentially hurt the generalization ability due to overfitting.

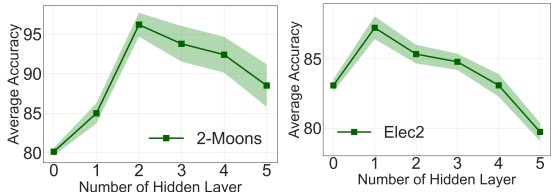

Figure 5: Sensitivity analysis on the number of layers of the generated neural network by DRAIN.

### 4.5 ABLATION STUDY

We further conduct an ablation study on three datasets to evaluate the effect of different components in DRAIN, and the results are exhibited in Table 2. Specifically, we remove the sequential learning model in DRAIN, and the resulted ablated model ✗ RNN corresponds to the offline baseline model. We also independently remove the skip connection module to let the sequential learning model uniformly acquire information from all previous domains, and the resulting model is named ✗ Skip.C.

Table 2: Ablation study. Comparison of performance between our method and two alternatives across two datasets for classification tasks and one datasets for regression tasks.

| Ablation | 2-Moons | Rot-MNIST | House |
|---|---|---|---|
| ✗ RNN | $22.4 \pm 4.6$ | $19.5 \pm 3.4$ | $11.0 \pm 0.36$ |
| ✗ Skip.C | $7.1 \pm 1.3$ | $10.3 \pm 1.7$ | $9.7 \pm 0.13$ |
| DRAIN | $\mathbf{3.2 \pm 1.2}$ | $\mathbf{7.5 \pm 1.1}$ | $\mathbf{9.3 \pm 0.14}$ |

As shown in the table, yet each component can effectively contribute to the overall model performance, modeling the temporal correlation between all domains by a sequential model can provide a rather larger performance gain. In addition, removing the skip connection in the sequential learning model would make DRAIN hard to capture the long-range temporal dependency among domains since long-range domain information could potentially be forgotten during the model learning.

## 5 CONCLUSION

We tackle the problem of temporal domain generalization by proposing a dynamic neural network framework. We build a Bayesian framework to model the concept drift and treat a neural network as a dynamic graph to capture the evolving pattern. We provide theoretical analyses of our proposed method, such as uncertainty and generalization error, and extensive empirical results to demonstrate the efficacy and efficiency of our method compared with state-of-the-art DA and DG methods.

ACKNOWLEDGEMENT

This work was supported by the National Science Foundation (NSF) Grant No. 1755850, No. 1841520, No. 2007716, No. 2007976, No. 1942594, No. 1907805, a Jeffress Memorial Trust Award, Amazon Research Award, NVIDIA GPU Grant, and Design Knowledge Company (subcontract number: 10827.002.120.04).

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

## A APPENDIX

### A.1 EXPERIMENTAL DETAILS

#### A.1.1 DATASET DETAILS

We expand upon the datasets used for our experiments in this section. We highlighted the sentence that describes the domain drift within each dataset.

- **Rotated 2 Moons:** This is a variant of the 2-entangled moons dataset, with a lower moon and an upper moon labeled 0 and 1 respectively. Each moon consists of 100 instances, and 10 domains are obtained by sampling 200 data points from the 2-Moons distribution, and rotating them counter-clockwise in units of $18°$. Domains 0 to 8 (both inclusive) are our training domains, and domain 9 is for testing. *Each domain is obtained by rotating the moons counter-clockwise in units of $18°$, hence the concept drift means the rotation of the moon-shape clusters.*

- **Rotated MNIST:** This is an adaptation of the popular MNIST digit dataset Deng (2012), where the task is to classify a digit from 0 to 9 given an image of the digit. We generate 5 domains by rotating the images in steps of 15 degrees. To generate the $i$-th domain, we sample 1,000 images from the MNIST dataset and rotate them counter-clockwise by $15 \times$ i degrees. We take the first four domains as train domains and the fifth domain as test. *Similar to 2-Moons, each domain here is generated by rotating the images of digits by $15°$, hence the concept drift means the rotation of the images.*

- **Online News Popularity:** This dataset Fernandes et al. (2015) summarizes a heterogeneous set of features about articles published by Mashable in a period of two years. The goal is to predict the number of shares in social networks (popularity). We split the dataset by time into 6 domains and use the first 5 for training. *The concept drift is reflected in the change of time, but previous works have proven Nasery et al. (2021) the concept drift is not strong.*

- **Shuttle:** This dataset provides about 58,000 data points for space shuttles in flight. The task is multiclass classification with a heavy class imbalance. The dataset was divided into 8 domains based on the time points associated with points, with times between 30-70 being the train domains and 70 -80 being the test domain.

- **Electrical Demand** This contains information about the demand of electricity in a particular province. The task is, again binary classification, to predict if the demand of electricity in each period (of 30 mins) was higher or lower than the average demand over the last day. We consider two weeks to be one time domain, and train on 29 domains while testing on domain 30. *Each domain is generated by considering the demand of electricity within certain two weeks, so the domain drift can be regarded as how the electricity demand is changing seasonally.*

- **House Prices Dataset:** This dataset has housing price data from 2013-2019. This is a regression task to predict the price of a house given the features. We treat each year as a separate domain, but also give information about the exact date of purchase to the models. We take data from the year 2019 to be test data and prior data as training. *Similar to Elec2, the concept drift in this dataset is how the housing price changed from 2013-2019 for a certain region.*

- **Appliances Energy Prediction:** This dataset Candanedo et al. (2017) is used to create regression models of appliances energy use in a low energy building. The data set is at 10 min for about 4.5 months in 2016, and we treat each half month as a single domain, resulting in 9 domains in total. The first 8 domains are used for training and the last one is for testing. *Similar to Elec2, the drift for this dataset corresponds to how the appliances energy usage changes in a low energy building over about 4.5 months in 2016.*

#### A.1.2 DETAILS OF COMPARISON METHODS

- **Practical Baseline.** *1). Offline:* this is a time-oblivious model that is trained using ERM on all the source domains. *2). LastDomain:* this is a time-oblivious model that is trained using ERM on the last source domains. *3). IncFinetune:* we bias the training towards more recent data by applying the Baseline method described above on the first time point and then, fine-tuning with a reduced learning rate on the subsequent time points in sequential manner. This baseline corresponds to the online model we defined in Definition 1.

- **Continuous Domain Adaptation Methods.** *1). CDOT:* this model transports most recent labeled examples $\mathcal{D}^T$ to the future using a learned coupling from past data, and trains a classifier on them.. *2). CIDA:* this method is representative of typical domain erasure methods applied to continuous domain adaptation problems. *3). Adagraph:* This method makes the batch norm parameters time-sensitive and smooths them using a given kernel.

- **Temporal Domain Generalization Method.** *1). GI:* this method proposes a training algorithm to encourage a model to learn functions which can extrapolate well to the near future by supervising the first order Taylor expansion of the learnt function.

### A.1.3 PARAMETER SETTING

We use Adam optimizer for all our experiments, and the learning rate for all datasets are uniformly set to be $1e - 4$. All experiments are conducted on a $64$-bit machine with $4$-core Intel Xeon W-2123 @ 3.60GHz, 32GB memory and NVIDIA Quadro RTX 5000. We set hyperparameters for each comparison method with respect to the recommendation in their original paper, and we specify the architecture as well as other details for each dataset's experiments as follows.

- **2-Moons.** The number of layers in the LSTM is set to be $10$, and the network architecture of $g_{\omega_t}$ consists of 2 hidden layers, with a dimension of 50 each. We use ReLU layer after each hidden layer and a Sigmoid layer after the output layer. The learning rate is set to be $1e - 4$.

- **Rot-MNIST.** The number of layers in the LSTM is set to be $10$, and the network architecture of $g_{\omega_t}$ consists of $2$ convolution layers with kernel shape $3 \times 3$, and each convolution layer is followed by a max pooling layer with kernel size $2$ and stride $= 2$. The latent representation is then transformed by two linear layers with dimensions $256$ and $10$. We use ReLU layer after each hidden layer and a Sigmoid layer after the output layer. The learning rate is set to be $1e - 3$.

- **ONP.** The number of layers in the LSTM is set to be $10$, and the network architecture of $g_{\omega_t}$ consists of 2 hidden layers with bias terms, and the dimensions of each layer are 20. We use ReLU layer after each hidden layer and a Sigmoid layer after the output layer. The learning rate is set to be $1e - 4$.

- **Shuttle.** The number of layers in the LSTM is set to be $5$, and the network architecture of $g_{\omega_t}$ consists of $3$ hidden layers with bias terms, and the dimensions of each layer are $128$. We use ReLU layer after each hidden layer and a Sigmoid layer after the output layer. The learning rate is set to be $5e - 5$.

- **Elec2.** The number of layers in the LSTM is set to be $10$, and the network architecture of $g_{\omega_t}$ consists of 2 hidden layers with bias terms, and the dimensions of each layer are 128. We use ReLU layer after each hidden layer and a Sigmoid layer after the output layer. The learning rate is set to be $5e - 5$.

- **House.** The number of layers in the LSTM is set to be $10$, and the network architecture of $g_{\omega_t}$ consists of 2 hidden layers with bias terms, and the dimensions of each layer are 128. We use ReLU layer after each hidden layer and no activation layer after the output layer. The learning rate is set to be $1e - 5$.

- **Appliance.** The number of layers in the LSTM is set to be $10$, and the network architecture of $g_{\omega_t}$ consists of 2 hidden layers with bias terms, and the dimensions of each layer are 128. We use ReLU layer after each hidden layer and no activation layer after the output layer. The learning rate is set to be $1e - 5$.

### A.1.4 TRAINING TIME ANALYSIS

Table 3: Comparison of training time (seconds) between our method and two baselines across two datasets for classification tasks and one datasets for regression tasks.

| Model | 2-Moons | Elec2 | Appliance |
|---|---|---|---|
| CIDA | $45.2 \pm 0.87$ | $154.3 \pm 2.1$ | $287.5 \pm 2.6$ |
| GI | $19.3 \pm 0.43$ | $136.4 \pm 1.9$ | $189.3 \pm 2.1$ |
| DRAIN (ours) | $\mathbf{15.4 \pm 0.37}$ | $\mathbf{99.2 \pm 1.3}$ | $\mathbf{170.3 \pm 1.8}$ |

We further conduct the model scalability analysis by comparing the running time of our proposed method with two other state-of-the-art baselines: GI and CIDA on three datasets (i.e., 2-Moons, Elec2, and Appliance). As shown in Table 3, our proposed method can generally achieve the shortest training time among the three methods. However, we notice that GI is relatively slower in the total running time due to the model pretraining and finetuning step, and the low efficiency in CIDA is due to the expensive computation cost for training GAN. Compared to these approaches, DRAIN only consists of one sequential learning model to address the data distribution drift in the end-to-end manner, which could achieve generally better performance while attaining its efficiency.

### A.1.5 SCALABILITY OF NUMBER OF DOMAINS

The time complexity of our framework with respect to the number of domains is linear (equivalent to the complexity of the recurrent neural network with respect to the input sequence length). The number of domains can only affect the total training time since we need to iteratively feed in a new domain to train the proposed recurrent model.

We conduct the following experiment to support our argument. We create synthetic datasets with 10, 100, and 1000 domains, each of which has two labels with 10 training instances. We follow the parameter setting in the 2-Moons dataset (the exact parameter setting can be found in Appendix A.3), and their runtime is demonstrated in the following table.

Table 4: Scalability of DRAIN for number of training domains.

| Number of domains | Running time |
| --- | --- |
| 10 | 2.66 |
| 100 | 28.51 |
| 1000 | 292.49 |

### A.1.6 IMPORTANT REMARKS

In this section, we provide some important remarks over the proposed DRAIN framework.

- Graph generation can handle large graphs and there are a number of existing works that can handle large graphs. Our model is a general framework that can choose different graph generation methods as needed.

- Neural networks are networks (i.e., graphs) of neurons, which have gained lots of research interest in recent years. Recent research (e.g., (You et al., 2020)) have found that the performance of neural network architectures is highly correlated with certain graph characteristics. In this work, we aim at characterizing the potential drift across the domains by optimizing and updating the graph structure of the neural network because optimizing the graph structure of a neural network has been proven to have a smaller search space and a more smooth optimization procedure than exhaustively searching over all possible connectivity patterns. Last but not least, our approach allows the entire neural network/model to change across time, which in turn maximizes our model's expressiveness.

### A.1.7 ENLARGED DECISION BOUNDARY FIGURES OF GI AND DRAIN

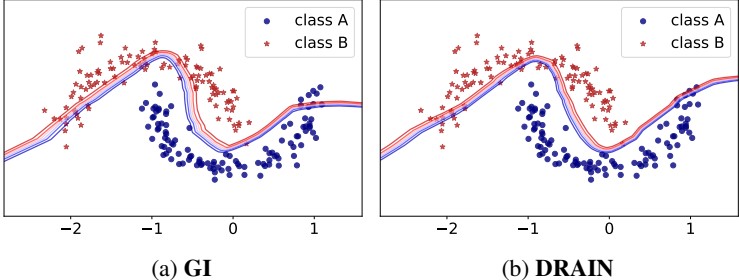

(a) **GI**                              (b) **DRAIN**

Figure 6: Comparison of the decision boundary on the future domain of 2-Moons dataset between the state-of-the-art model - GI and the proposed model - DRAIN.

Figure 6 is a direct comparison of decision boundaries predicted by the state-of-the-art method GI (Figure 6a) and the proposed method DRAIN (Figure 6b). As can be seen from the figure, the decision boundary predicted by DRAIN can consistently classify two classes with a few exceptions. the decision boundary predicted by GI has less confidence (i.e., wider band) in predicting middle points, and a few errors are also made in predicting points on the right side.

## A.2 OVERALL GENERATION PROCESS

We summarize the detailed forward propagation of DRAIN as below:

$$a_1 = 0, \ m_1 = G_0(z), \ z \sim \mathcal{N}(0, 1)$$
$$a_1 = G_\eta(\omega_1), \ \omega_1 \sim F_\xi(h_1), (m_1, h_1) = f_\theta(m_0, a_0)$$
$$\cdots$$
$$a_1 = G_\eta(\omega_1), \ \omega_1 \sim G_o(h_1), (m_1, h_1) = f_\theta(m_1, a_1)$$
$$a_2 = G_\eta(\omega_2), \ \omega_2 = \Phi(\omega_2, \{\omega_1\}), \ \omega_2 = F_\xi(h_1), (m_2, h_2) = f_\theta(m_1, a_1)$$
$$\cdots$$
$$a_2 = G_\eta(\omega_2), \ \omega_2 = \Phi(\omega_2, \{\omega_1\}), \ \omega_2 = F_\xi(h_1), (m_2, h_2) = f_\theta(m_2, a_2)$$
$$\cdots$$
$$a_t = G_\eta(\omega_2), \ \omega_t = \Phi(\omega_t, \{\omega_{t-\tau:t-1}\}), \ \omega_2 = F_\xi(h_1), (m_t, h_t) = f_\theta(m_t, a_t),$$

where each $a_i$ denotes the input of $f_\theta$. In this work, we utilize LSTM as the recurrent architecture, and $f_\theta$ becomes a single LSTM unit. To initialize the whole generative process, we take a random noise $z$ as input for the first domain $\mathcal{D}_1$, which is drawn from a standard Gaussian distribution. The initial memory state $m_1$ is also transformed from $z$ by an initial encoding function $G_0(\cdot)$.

## A.3 THEORY PROOF

In this section, we provide the formal proof for Theorem 1 and Theorem 2 in our main context.

### A.3.1 PROOF FOR THEOREM 1

*Proof.* By definition of the predictive distribution,

$$
\begin{aligned}
P(\hat{y}|\mathbf{x}_{T+1}, \mathcal{D}_{1:T}) &= \int P(\hat{y}|\mathbf{x}_{T+1}, \omega_{T+1}) P(\omega_{T+1}|\mathcal{D}_{1:T}) d\omega_{T+1} \\
&= \int P(\hat{y}|\mathbf{x}_{T+1}, \omega_{T+1}) P(\omega_{T+1}|\omega_{1:T}) P(\omega_{1:T}|\mathcal{D}_{1:T}) d\omega_{1:T+1}
\end{aligned}
\tag{5}
$$

Our goal is to prove that the variance of this predictive distribution for our proposed method, online baseline and offline baseline follows the inequality as in Theorem1.

**Ours v.s. Online Baseline**

Here we prove that $\text{Var}(\mathcal{M}_{\text{ours}}) < \text{Var}(\mathcal{M}_{\text{on}})$.

Notice that the first term on the right hand side of Eq. 5, namely $P(\hat{y}|\mathbf{x}_{T+1}, \omega_{T+1})$, corresponds to deployment of the model with parameter $\omega_{T+1}$ on the future domain $\mathcal{D}_{T+1}$, hence the variance of $P(\hat{y}|\mathbf{x}_{T+1}, \omega_{T+1})$ only depends on the noise or randomness coming from $\mathbf{x}_{T+1}$ as long as $\omega_{T+1}$ is given. In other words, the uncertainty coming from $P(\hat{y}|\mathbf{x}_{T+1}, \omega_{T+1})$ can be cancelled for both methods since we are considering the same set of domains. Now the problem reduces to prove that the variance of the second and third terms on the right hand side of Eq. 5 for our model is smaller than those for the online baseline.

Notice that

$$
\begin{aligned}
&P(\omega_{1:T}|\mathcal{D}_{1:T}) \\
&= \int_\Theta P(\omega_1|\mathcal{D}_1) \cdot P(\omega_2|\omega_1, \mathcal{D}_2, \theta_0) \cdot P(\theta_1|\omega_1, \omega_2, \theta_0) \cdot P(\omega_3|\omega_2, \mathcal{D}_3, \theta_1) \cdot P(\theta_2|\omega_2, \omega_3, \theta_1) \\
&\quad \cdots P(\omega_T|\omega_{T-1}, \mathcal{D}_T, \theta_{T-2}) \cdot P(\theta_{T-1}|\omega_{T-1}, \omega_T, \theta_{T-2}) d\theta_{0:T-1},
\end{aligned}
\tag{6}
$$

where $\theta$ is the parameter of the parameterized function to approximate the ground-truth drift of $\omega$, as defined in Assumption 1. For example, $P(\omega_1|\mathcal{D}_1)$ denotes that we train the model on the very first

domain and $P(\omega_2|\omega_1, \mathcal{D}_2, \theta_0)$ denotes that we continue to train the model on the second domain but with initialization of $\omega_2$ as $q_{\theta_0}(\omega_1)$ where $\omega_1$ is learned from the previous domain and $q_{\theta_0}$ is trying to capture the conditional probability or drift between $\omega_2$ and $\omega_1$, i.e., $P(\omega_2|\omega_1)$. In our Bayesian framework, we treat $q_\theta$ as a learnable function (e.g., LSTM unit in our proposed method) and we use subscript of $\theta$ to differentiate the status of $\theta$ after the training on each domain. In other words, $q_\theta$ will be updated after the training on each domain (at least for our method). Notice that $\theta_0$ always denotes the parameter initialization as in Assumption 1.

By Bayes' rule, we have:

$$P(\omega_{t+1}|\omega_t, \mathcal{D}_{t+1}, \theta_{t-1}) \propto \underbrace{P(q_{\theta_{t-1}}(\omega_t))}_{\text{prior on } \omega_{t+1}} \cdot \underbrace{P(\mathcal{D}_{t+1}|\omega_{t+1})}_{\text{likelihood}}, \tag{7}$$

where $P(q_{\theta_{t-1}}(\omega_t))$ can be regarded as the prior of $\omega_{t+1}$ because as we mentioned $q_{\theta_{t-1}}$ denotes the initialization of $\omega_{t+1}$ before we train the model on domain $\mathcal{D}_{t+1}$, and $P(\mathcal{D}_{t+1}|\omega_{t+1})$ corresponds to the likelihood of training $\omega_{t+1}$ on $\mathcal{D}_{t+1}$. In addition,

$$
\begin{aligned}
P(\theta_t|\omega_t, \omega_{t+1}, \theta_{t-1}) &\propto P(\theta_{t-1}) \cdot P(\omega_t, \omega_{t+1}|\theta_t) \\
&\propto P(\theta_{t-2}) \cdot P(\omega_{t-1}, \omega_t|\theta_{t-1}) \cdot P(\omega_t, \omega_{t+1}|\theta_t) \\
&\cdots \\
&\propto \underbrace{P(\theta_0)}_{\text{prior on } \theta} \cdot \underbrace{\prod_{i=1}^{t} P(\omega_i, \omega_{i+1}|\theta_i)}_{\text{likelihood}},
\end{aligned}
\tag{8}
$$

for any $t = 1, 2, 3, \cdots, T-1$. In the equation above, this time the prior is over parameter $\theta$ and $\omega_i$, $\omega_{i+1}$ can be regarded as the "training data" for $\theta_i$.

For the online baseline, since it only keeps one-step finetuning of the model and does not learn how $\omega_t$ evolves, the $\theta_t$ for the online baseline is always equal to the prior, i.e. $\theta_t = \theta_0$. In other words, $P(q_{\theta_{t-1}}(\omega_t)) = P(q_{\theta_0}(\omega_t))$ and $P(\theta_t|\omega_t, \omega_{t+1}, \theta_{t-1}) = P(\theta_0)$, $\forall t$ for the online baseline.

Since we follow the standard routine and assume all distributions are Gaussian, by Bayesian Theorem, we know that the posterior distribution always has variance smaller than the prior distribution under the expectation, i.e.,

$$\mathbb{E}\big[Var(\theta_t|\omega_t, \omega_{t+1}, \theta_{t-1})\big] < Var(\theta_0), \tag{9}$$

which proves that our method has smaller variance in terms of Eq. 8. On the other hand, since the second term on the right hand side of Eq. 7 is the same for both methods, and for the first term $P(q_{\theta_{t-1}}(\omega_t))$, by our Assumption 1 we know that for baseline $\Pr(q_{\theta_{t-1}}(\omega_t)) = \Pr(q_{\theta_0}(\omega_t))$ so the variance is basically $\sigma_{\theta_0}$. For our method, after each training step across a new domain our $\theta$ will get updated and achieve smaller variance (because of posterior variance of Gaussian) so we also prove that our method has smaller variance in terms of Eq. 7. Two parts combined prove that our method has smaller variance in the third term of Eq. 5, namely $P(\omega_{1:T}|\mathcal{D}_{1:T})$.

The last step is to compare the variance from the second term in Eq. 5, namely $P(\omega_{T+1}|\omega_{1:T})$. For online baseline, basically it uses the parameter from the last training domain, i.e., $\omega_T$ as the final model on the future domain, i.e., $P(\omega_{T+1}|\omega_{1:T}) = P(q_{\theta_0}(\omega_T))$.

On the other hand, for our method we have $P(\omega_{T+1}|\omega_{1:T}) = P(q_{\theta_{T-1}}(\omega_T))$ which has smaller variance due to the posterior variance of Gaussian.

All together we finish the proof for $Var(\mathcal{M}_{\text{ours}}) < Var(\mathcal{M}_{\text{on}})$.

**Online Baseline v.s. Offline Baseline**

This case is simpler to prove. Again, the first term on the right hand side of Eq 5, namely $P(\hat{y}|\mathbf{x}_{T+1}, \omega_{T+1})$ can be cancelled in this case. Moreover, the second term, namely $P(\omega_{T+1}|\omega_{1:T})$ has the same variance for both baselines, i.e., $Var(P(\omega_{T+1}|\omega_{1:T})) = Var(P(q_{\theta_0}(\omega_T))) = \sigma_{\theta_0}$. This makes sense since two baselines do not learn the drift and the uncertainty in predicting $\omega_{T+1}$ based on $\omega_T$ is always the same as the prior distribution of $\theta_0$.

Hence, it suffices to compare the uncertainty of the last term of Eq. 5, namely $P(\omega_{1:T}|\mathcal{D}_{1:T})$. Recall

$$
\begin{aligned}
\mathcal{M}_{\text{on}}: \quad &\omega_{t+1} = \text{argmax}_{\omega_{t+1}} P(\omega_{t+1}|\omega_t, \mathcal{D}_{t+1}) \\
\mathcal{M}_{\text{off}}: \quad &\omega_{t+1} = \text{argmax}_{\omega_{t+1}} P(\omega_{t+1}|\mathcal{D}_{1:t+1})
\end{aligned}
\tag{10}
$$

For offline baseline, we are using all dataset so far, namely $\mathcal{D}_{1:t+1}$ to train the model while the online baseline only uses $\mathcal{D}_{t+1}$. Since we are considering domain generalization with temporal concept drift, i.e., for each $i \neq j$ we have $\mathcal{D}_i \neq \mathcal{D}_j$ (otherwise we merge them), the randomness of $\bigcup_{i=1}^{t+1} \mathcal{D}_i$ is at least as large as that of $\mathcal{D}_{t+1}$ alone, i.e., $\mathrm{Var}(\bigcup_{i=1}^{t+1} \mathcal{D}_i) \geq \mathrm{Var}(\mathcal{D}_{t+1})$.

To prove this, let's consider the case of two domains $\mathcal{D}_1$ and $\mathcal{D}_2$ without loss of generality. Also, assume the sample size for both domains are equal. By definition of variance, we have

$$\mathrm{Var}(\mathcal{D}_1) = \frac{\sum_{i=1}^{n}(x_{1,i} - \mu_1)^2}{n}, \quad \mathrm{Var}(\mathcal{D}_2) = \frac{\sum_{i=1}^{n}(x_{2,i} - \mu_2)^2}{n}, \tag{11}$$

while

$$\mathrm{Var}(\mathcal{D}_1 \cup \mathcal{D}_2) = \frac{\sum_{i=1}^{n}(x_{1,i} - \frac{\mu_1+\mu_2}{2})^2 + \sum_{i=1}^{n}(x_{2,i} - \frac{\mu_1+\mu_2}{2})^2}{2n}, \tag{12}$$

where $\mu_1$ and $\mu_2$ is the sample mean for each domain, respectively and $n$ denotes the sample size. Hence,

$$
\begin{aligned}
&\mathrm{Var}(\mathcal{D}_1 \cup \mathcal{D}_2) - \frac{1}{2}\Big(\mathrm{Var}(\mathcal{D}_1) + \mathrm{Var}(\mathcal{D}_2)\Big) \\
=& \frac{\sum_{i=1}^{n}(x_{1,i} - \frac{\mu_1+\mu_2}{2})^2 + \sum_{i=1}^{n}(x_{2,i} - \frac{\mu_1+\mu_2}{2})^2}{2n} - \frac{\sum_{i=1}^{n}(x_{1,i} - \mu_1)^2 + \sum_{i=1}^{n}(x_{2,i} - \mu_2)^2}{2n} \\
\propto& \sum_{i=1}^{n}(x_{1,i} - \frac{\mu_1+\mu_2}{2})^2 + \sum_{i=1}^{n}(x_{2,i} - \frac{\mu_1+\mu_2}{2})^2 - \Big(\sum_{i=1}^{n}(x_{1,i}-\mu_1)^2 + \sum_{i=1}^{n}(x_{2,i}-\mu_2)^2\Big) \\
=& \sum_{i=1}^{n}\Big((x_{1,i} - \frac{\mu_1+\mu_2}{2})^2 + (x_{2,i} - \frac{\mu_1+\mu_2}{2})^2 - \big[(x_{1,i}-\mu_1)^2 + (x_{2,i}-\mu_2)^2\big]\Big) \\
=& \sum_{i=1}^{n}\Big(-(\mu_1+\mu_2)x_{1,i} - (\mu_1+\mu_2)x_{2,i} + 2\mu_1 x_{1,i} + 2\mu_2 x_{2,i} + \frac{1}{2}(\mu_1+\mu_2)^2 - \mu_1^2 - \mu_2^2\Big) \\
=& \sum_{i=1}^{n}\Big((\mu_1-\mu_2)x_{1,i} - (\mu_1-\mu_2)x_{2,i} - \frac{1}{2}(\mu_1-\mu_2)^2\Big) \\
=& \sum_{i=1}^{n}\Big((\mu_1-\mu_2)(x_{1,i}-x_{2,i}) - \frac{1}{2}(\mu_1-\mu_2)^2\Big) \\
=& \sum_{i=1}^{n}\Big((\mu_1-\mu_2)^2 - \frac{1}{2}(\mu_1-\mu_2)^2\Big) \\
=& \sum_{i=1}^{n}\frac{(\mu_1-\mu_2)^2}{2} \geq 0,
\end{aligned}
\tag{13}
$$

where the equation from the third last row to the second last row is under expectation as $\mathbb{E}\big[(x_{1,i}-x_{2,i})\big] = \mu_1 - \mu_2$. Since we assume $\mathrm{Var}(\mathcal{D}_1) = \mathrm{Var}(\mathcal{D}_2)$, we finish the proof that $\mathrm{Var}(\mathcal{D}_1 \cup \mathcal{D}_2) \geq \mathrm{Var}(\mathcal{D}_2)$. One can generalize this conclusion onto three or more domains.

Finally, combining $\mathrm{Var}(\bigcup_{i=1}^{t+1} \mathcal{D}_i)$ is at least as large as that of $\mathrm{Var}(\mathcal{D}_{t+1})$ with Bayes' rule, one can finish the proof.

$\square$

### A.3.2 PROOF OF THEOREM 2

*Proof.* We finish our proof in **two steps**. First we prove that the generalization error of our method is smaller than that of the online baseline.

By definition, we know that

$$err := \ell\Big(\mathbb{E}\big[P(\hat{y}_{T+1}|\mathbf{x}_{T+1}, \mathcal{D}_{1:T})\big], y_{T+1}\Big) = \ell\Big(g_{\omega_{T+1}}(\mathbf{x}_{T+1}), y_{T+1}\Big), \tag{14}$$

where $g_\omega$ denotes the target neural network with parameter $\omega$, and $\omega_{T+1}$ denotes the parameter status on the $(T+1)$-th domain (i.e., future domain).

For online baseline, since it does not consider the temporal information, the parameters on the future domain will be the same as the parameters after the training on the last source domain, i.e, for online baseline we have $\omega_{T+1} = \omega_T$.

For our method, we have $\omega_{T+1} = q_{\theta_T}(\omega_T)$, where $q_\theta$ is the recurrent structure and $\theta_T$ denotes the parameter status of the recurrent structure after training on the first $T$ domains. In other words, our method utilizes the recurrent structure to generate the model parameters on the next domain. Now it suffices to show that

$$\ell\Big(g_{\omega_{T+1}}(\mathbf{x}_{T+1}), y_{T+1}\Big) < \ell\Big(g_{\omega_T}(\mathbf{x}_{T+1}), y_{T+1}\Big), \tag{15}$$

where $\omega_{T+1} = q_{\theta_T}(\omega_T)$. Here, the LHS and RHS above corresponds to the generalization error of our method and the online baseline, respectively.

Recall that $q_\theta$ represents the LSTM unit in our case, and we train the LSTM unit to approximate the transition probability $P(\omega_{t+1}|\omega_t)$, i.e., how neural network $g$'s parameter distribution changes over time. From a probabilistic point of view, during training of the LSTM unit $q_\theta$, we basically minimize the empirical loss which is equivalent to

$$\min_\theta D_{\mathrm{KL}}\Big(q_\theta \big\| P(\omega_{t+1}|\omega_t)\Big), \quad t = 1, 2, \cdots, T-1. \tag{16}$$

As mentioned in Assumption 1, we denote $\theta_0$ as the initialization of $q_\theta$. On the other hand, after $T-1$ times of training over the LSTM unit on the $T$ source domains, $\theta$ will converge to an optima denoted as $\theta^*$. Hence, the model parameter $\omega_{T+1}$ generated by the converged LSTM unit for sure will be closer to the ground truth than that generated by the random initialized LSTM unit, i.e.,

$$\big\| q_{\theta_T}(\omega_T) - q_{\theta^*}(\omega_T) \big\| < \big\| q_{\theta_0}(\omega_T) - q_{\theta^*}(\omega_T) \big\|. \tag{17}$$

By Lipschitz continuity of $g_\omega$ over the parameter $\omega$, we have

$$L_{lower} \cdot \big\| \omega - \omega' \big\| < \big\| g_\omega(x) - g_{\omega'}(x) \big\| < L_{upper} \cdot \big\| \omega - \omega' \big\|, \quad \forall\, x \in \mathcal{X}, \tag{18}$$

where $\mathcal{X}$ is defined as the input space of neural network $g_\omega$. Bubeck et al. (2021) proved that the Lipschitz constant actually can have a lower bound for a neural network.

Denote $\omega^* = q_{\theta^*}(\omega_T)$, i.e., the optimal parameter for the target neural network $g$ on the future domain. Then, it directly follows Eq. 18 that

$$\begin{aligned} \big\| g_{\omega_T}(x) - g_{\omega^*}(x) \big\| &> L_{lower} \cdot \big\| \omega_T - \omega^* \big\|, \\ \big\| g_{\omega_{T+1}}(x) - g_{\omega^*}(x) \big\| &< L_{upper} \cdot \big\| \omega_{T+1} - \omega^* \big\|. \end{aligned} \tag{19}$$

Denote $r = \|\omega_T - \omega^*\| / \|\omega_{T+1} - \omega^*\|$. Since neural network $g_\omega$ is a continuous function of $\omega$, there always exists a constant $\delta > 0$ such that, within the sphere centering at $\omega^*$ with radius $\delta$, namely $\mathcal{S}(\omega^*, \delta)$, the local lower and upper bound for the Lipschitz constant of $g_\omega$ could satisfy $L_{upper}/L_{lower} < r$. The reason behind this is, as $\delta$ approaches 0, due to the continuity of $g_\omega$, the upper bound and lower bound of Lipschitz constant within $\mathcal{S}(\omega^*, \delta)$ will become closer and finally identical, i.e., $\lim_{\delta \to 0^+} L_{upper}/L_{lower} = 1$. On the other hand, by Eq. 17 we know that $r$ is always greater than 1, so it is always possible to find a $\delta$ to satisfy the above condition. As a result,

$$\frac{L_{upper}}{L_{lower}} \cdot \frac{\|\omega_{T+1} - \omega^*\|}{\|\omega_T - \omega^*\|} < 1 \iff L_{upper} \cdot \big\| \omega_{T+1} - \omega^* \big\| < L_{lower} \cdot \big\| \omega_T - \omega^* \big\|. \tag{20}$$

Hence, $\big\| g_{\omega_{T+1}}(x) - g_{\omega^*}(x) \big\| < \big\| g_{\omega_T}(x) - g_{\omega^*}(x) \big\|$. Since $g_{\omega^*}(x)$ is the optimal neural network on the future domain, $g_{\omega^*}(\mathbf{x}_{T+1})$ should achieve the lowest loss as defined in Eq. 15. Combined everything above together finishes the first step of our proof.

The second step of our proof is for the comparison between two baselines. We consider the case where the drift of $\omega_t$ is monotonic but our proof can be generalized to other cases easily.

As can be shown,

$$\begin{aligned} \text{Online baseline:} \quad & \mathbb{E}[P(\omega_{T+1}|\omega_{1:T})] = \mathbb{E}[q_{\theta_0}(\omega_T)] = \omega_T, \\ \text{Offline baseline:} \quad & \mathbb{E}[P(\omega_{T+1}|\omega_{1:T})] = \mathbb{E}[q_{\theta_0}(\bar{\omega})] = \mathbb{E}[P(\omega|\mathcal{D}_{1:T})]. \end{aligned} \tag{21}$$

If we denote a distance function over the domains, as $d$, we assume that $d(\mathcal{D}_{t+1}, \mathcal{D}_{T+1}) < d(\mathcal{D}_t, \mathcal{D}_{T+1})$. By the monotonic assumption, the distribution of each $\mathcal{D}_{1:T}$ is changing along a certain direction. Hence, among them $\mathcal{D}_T$ has the distribution most close to that of $\mathcal{D}_{T+1}$. In other words, the online baseline finetunes the model so its $\omega_T$ is leaning towards the last domain while the offline baseline is using the averaged domains to train the model, which finishes the proof.

$\square$

