# OpenReview forum: "Temporal Domain Generalization with Drift-Aware Dynamic Neural Networks"
_ICLR.cc/2023/Conference — ICLR 2023 notable top 5%_

### Official Review · Reviewer_NDvd · 2022-10-17

**Confidence:** 3
**Correctness:** 3
**Technical Novelty And Significance:** 4
**Empirical Novelty And Significance:** 3
**Recommendation:** 6

**Clarity, Quality, Novelty And Reproducibility:**

Overall, the paper is dense but remains quite clear. The authors tackle an important problem with an intuitive and elegant novel solution.

Miscellaneous:
- Is “Incfinetune” the same as “Temporal” in 4c ?
- Do you have an intuition why “CIDA utilizes the adversarial training technique to solve the domain adaptation, yet the predicted decision boundary in Figure 4e is less stable than other state-of-the-art methods due to its model complexity.” is not reflected in the variance of the performance in Table 1?
- Table 2 line one does not correspond to the performance of IncFinetune but offline (so I think there is a typo somewhere)


**Strength And Weaknesses:**

The paper approaches the important problem of temporal concept drift with an elegant solution. The draft is quite dense but remains clear. The experimentation is thorough with useful ablation studies and interesting discussions and limitations.

The theoretical results are interesting and show the importance of modelling the concept drift, but do not quite show why the proposed approach should outperform the other state-of-the-art strategies. Importantly, two key claims are unclear:
- Why should the variance of the union of domains always be larger than the last domain? For instance, if each domain at time $t$ is a gaussian with mean $0$ and variance $t$ (the variance increases with time, meaning that the union has a lower variance than the last observed distribution)
- Why does the posterior always have a lower variance than the prior? I think this claim only stands in expectation.
Finally, the formalisation of the concept drift under which the methodology would perform better would be particularly valuable.

**Summary Of The Paper:**

This paper tackles the issue of temporal drift. The proposed methodology consists of a recurrent neural network modelling the parameters of the classification/regression model, assuming constant topology of this later. This recurrent model aims to encapsulate the latent change in parameters through time. The parameters of the network at a new time point are then drawn from this latent state that encodes the concept drift. The model is compared to multiple strategies on a set of both synthetic and real datasets for both regression and classification tasks.


**Summary Of The Review:**

The paper studies an interesting problem and proposes a novel and efficient solution. However, the theoretical results do not strengthen considerably the paper and I am not sure of their correctness.

---

> ### Author Response · Authors · 2022-11-13
> **Author Response to Reviewer NDvd**
>
> Dear Reviewer,
>
> Thank you for your appreciation of the novelty of our method and the quality of our presentation. We have provided more details and explanations **both directly in the rebuttal and in our re-uploaded paper**, to better answer your questions. Please refer to our response below for detail.
>
> > *“Is “Incfinetune” the same as “Temporal” in 4c ?”*
>
> **A1:** **Yes**, the “Temporal" is the same as the “IncFinetune" method. We have revised this typo in **Figure 4** in our re-uploaded version. Thanks.
>
> > *“Do you have an intuition why “CIDA utilizes the adversarial training technique to solve the domain adaptation, yet the predicted decision boundary in Figure 4e is less stable than other state-of-the-art methods due to its model complexity.” is not reflected in the variance of the performance in Table 1?”*
>
> **A2:** The decision boundaries as shown in Figure 3 and Figure 4 are based on the **2-Moons dataset**. Actually, the stability of CIDA's decision boundary in Figure 4e **is reflected by** the variance of performance in the “2-Moons" column in Table 1. Specifically, the **standard deviation** of CIDA on 2-Moons is $1.6$, which is larger than GI's and DRAIN's standard deviation, i.e., $1.4$ and $1.2$, respectively. This pattern is reflected in Figure 4, as the decision boundary for CIDA is wider than those of GI and DRAIN and a wider decision boundary means a larger standard deviation.
>
> > *“Table 2 line one does not correspond to the performance of IncFinetune but offline (so I think there is a typo somewhere)”*
>
> **A3:** This is a typo. The first row of Table 2 corresponds to the offline method. Thank you for pointing it out and we have revised this in our re-uploaded version in **Table 2**.
>
> > *“Why should the variance of the union of domains always be larger than the last domain? For instance, if each domain at a time  is a gaussian with mean  and variance  (the variance increases with time, meaning that the union has a lower variance than the last observed distribution)”*
>
> **A4:** We have added additional proof for $Var(\cup_{i=1}^{T+1}D_i) \geq Var(D_{T+1})$ if the variance of each domain is equal, which has been clarified in our Theorem 1 in the main text. Please refer to **page 18 of Proof 1** in our re-uploaded version.
>
> > *“Why does the posterior always have a lower variance than the prior? I think this claim only stands in expectation. Finally, the formalization of the concept drift under which the methodology would perform better would be particularly valuable.”*
>
> **A5.** Yes, we are considering the claim “posterior has a lower variance than the prior" under the expectation. We have clarified this part with better explanations in our **Proof 1** in the re-uploaded version.
>
> We do not think there exists a specific formalization of the concept drift under which our method will perform better. However, a general answer to this question will be that our method can work well when the concept drift of model parameter distribution **is learnable by the recurrent structure**. In other words, the parametric function $q_{\theta}$ can be trained to approximate the unknown probability $P(\omega_{t+1} | \omega_t)$. If the temporal drift is extremely random or chaotic such that it cannot be captured by $q_{\theta}$, our method may fail in such cases. By the way, the ONP dataset in Table 1 is a good example where the offline baseline achieves the best performance due to the absence of temporal drift within the data.

---

> > ### Comment · Reviewer_NDvd · 2022-12-05
> > **Response**
> >
> > Thank you for your answers, however, my main concern remains: 'The theoretical results are interesting and show the importance of modelling the concept drift, but do not quite show why the proposed approach should outperform the other state-of-the-art strategies.'
> > Note that the updated proof 2 would benefit to be a recurrence.

---

> > > ### Author Response · Authors · 2022-12-06
> > > **Follow-up Response from Author**
> > >
> > > Dear Reviewer,
> > >
> > > First, we sincerely appreciate that our theoretical results are interesting and important to you. Our proposed method theoretically enjoys two advantages over the SOTA methods: **1).** Our proposed method allows the entire neural network to adapt over time dynamically, which achieves the maximal expressiveness compared with e.g., CIDA which just takes time as an additional input feature, or GI which only adapts partial parameters of the neural network over time. **2).** Our proposed method captures the temporal drift in a fully end-to-end and adaptive manner, while other SOTA methods such as GI heavily rely on heuristics to regularize the first-order gradients. Note that the two theoretical advantages of our proposed method over the SOTA methods are also **empirically validated** in our experiments. Specifically, Table 1 quantitatively demonstrates the superior performance of our method over other SOTA methods on six real-world temporal DG benchmarks. In addition, Figures 3-4 further provide qualitative analyses that better validate the performance gain of our method on the 2-Moons dataset.
> > >
> > > Hope our comments could address your concern. Thank you.

---

### Official Review · Reviewer_WCLC · 2022-10-24

**Confidence:** 4
**Correctness:** 4
**Technical Novelty And Significance:** 3
**Empirical Novelty And Significance:** 3
**Recommendation:** 8

**Clarity, Quality, Novelty And Reproducibility:**


[originality] As far as I know, this paper delivered the first attempt to consider neural networks as dynamic graphs over a recurrent structure to capture the temporal drift of model parameter distributions. I found the proposed method interesting and intuitive, and believe it is non-trivial in tackling the problem of temporal domain generalization.

[quality] I found the proposed method theoretically and empirically sound. Extensive theoretical analyses are shown for both prediction error and uncertainty quantification.  Experiment results are quite comprehensive and demonstrate the proposed method’s effectiveness. Furthermore, the visualization of decision boundaries provided deep insights beyond the values in the tables.

[clarity] In general, I found this paper well-written and easy to follow.

[Reproducibility] Source code and formal proofs of this paper are provided by the authors. As far as I checked, I did not find mathematical mistakes in the proof.


**Details Of Ethics Concerns:**

I did not find any ethics concerns in this paper.

**Strength And Weaknesses:**

Strengths:

1.	The originality of this paper is great. The problem is clearly defined.

2.	The topic of temporal domain generalization is interesting and prompt.

3.	The proposed method is both theoretically and technically sound to me.

4.	How the authors capture the temporal drift of model parameter distributions by considering the dynamic graph over a recurrent structure is intuitive. This paper makes a non-trivial exploration of the temporal domain generalization problem.

5.	Extensive theoretical and empirical results round up the good work.

Weaknesses:

1.	Could the authors elaborate more on the reason of using skip-connection in Sec 3.2.4? In other words, why providing the LSTM unit with previous domain’s information will mitigate the catastrophic forgetting on the future domain?

2.	Some details are not clear enough. For example, what is the shape of the parameter $\omega$ into the recurrent unit? In addition, what initialization is used for the model parameters in this paper?


**Summary Of The Paper:**

This paper considers the problem of domain generalization where the domain index is across time. The authors proposed a Bayesian framework to explicitly model the concept drift over time, i.e., predicting the updated model on the future domain based on historical domains. A recurrent neural network is employed to learn the evolving dynamics of model parameter distribution based on how data distribution evolves across the domains. To achieve maximal expressiveness, the authors proposed to model the neural network as a dynamic graph which allows all connections between the neurons to adapt temporally. The authors also introduced a technique based on skip-connection to mitigate catastrophic forgetting. Theoretical analyses showed smaller margin of error and uncertainty in prediction of the proposed method compared with two baselines. The proposed method achieved excellent performances on several classification and regression datasets compared with several state-of-the-art domain generalization methods.

**Summary Of The Review:**

Overall, this paper makes a non-trivial and meaningful exploration of temporal domain generalization. Both theoretical and empirical results are provided with great performance. The paper is well-written and quite easy to follow. Hence, I would recommend accept of this paper. I am open to hearing from the authors if I made any misunderstandings.

---

> ### Author Response · Authors · 2022-11-13
> **Author Response to Reviewer WCLC**
>
> Dear Reviewer,
>
> We are happy to see that you found our method interesting and sound. Please refer to our response below for detail.
>
> > *“Could the authors elaborate more on the reason for using skip-connection in Sec 3.2.4? In other words, why providing the LSTM unit with the previous domain’s information will mitigate the catastrophic forgetting on the future domain?”*
>
> **A1:** Because there are temporal dependencies between data distributions in consecutive time points, which are reflected as the existence of both **time-invariant** patterns and **time-variant** patterns across consecutive time points. For example, some input features may constantly be much more important (i.e., has larger weights) than some other input features, which are time-invariant. But the degree of how much those “more important” features are more important than those “less important” features could gradually evolve with small changes over time, which is time-variant. In all, the skip connection can help explain the **time-invariant** patterns, so the recurrent unit could better focus on the remaining time-variant patterns.
>
> We will add the explanations above to our paper if given more space after acceptance. Thanks.
>
> > *“Some details are not clear enough. For example, what is the shape of the parameter $\omega$ in the recurrent unit? In addition, what initialization is used for the model parameters in this paper?”*
>
> **A2:** **For your first question**: the parameter $\omega$ is flattened to a one-dimension vector by the decoding function and fed into the recurrent unit for the iterative training purpose.
> **For your second question**: we initialize the model parameters to follow the standard normal distribution.

---

### Official Review · Reviewer_vJXs · 2022-10-26

**Confidence:** 3
**Correctness:** 3
**Technical Novelty And Significance:** 3
**Empirical Novelty And Significance:** Not applicable
**Recommendation:** 6

**Clarity, Quality, Novelty And Reproducibility:**

Few remarks on clarity:

1. Authors use special type of integral, without fully describing what kind of integration they really do.
.e.g. what they write in eq 4. is in standard mathematical literature known as Line/Contour integral.
If that is the case, which parametrization are you using? Please provide more details on the integration part.

2. Can you provide empirical evidences that your Theorem 2 holds

3. Regarding assumption 1 for the concept drift is Gaussian. Can you try to quantify this empirically?

4. Proof 2
- Eq 16 you add one more assumption on the expected L2 error for your model. Why is this assumption not in the statement of the Therem?
- Can you clarify part on the mild assumptions for the Lipschitz continuity?



**Strength And Weaknesses:**

Strength:
- Interesting problem setting
- Interesting approach with solid results.

Weaknesses:
- Paper does not provide enough information to understand the full details (e.g. see comments on integrals and proofs)
- Proofs and theorems are not fully clear


**Summary Of The Paper:**

The Paper studies temporal domain generalization with dynamic neural networks.
For a discrete set of datasets: D1, ..., D_T at different timestamps: t_1, ..., t_T, goal is to build a model that captures concept drift.
In particular, model learns how to predicts parameters of NN for timestamp T+1 for the mapping X_{T+1}-> T_{T+1}.

Authors, propose a Bayesian probabilistic framework, that is based on the law of total probability where integration over inference part P(w_{T+1}| w{1:T}, D{1:T}) and training part is done P(w{1:T}| D_{1:T}), in the end2end manner.

The authors provide two main theorems about uncertainty quantification and generalization error.

Authors, show on on 5 datasets for classification and 2 dtasets for regressions that they have very competitive performance w.r.t. baselines: offline, lastdomain, CDOT and others.





**Summary Of The Review:**

Interesting paper but needs more clarity w.r.t. theoretical guarantees.

---

> ### Author Response · Authors · 2022-11-13
> **Author Response to Reviewer vJXs**
>
> Dear Reviewer,
>
> We sincerely appreciate that you found our paper and method interesting with solid results. We have provided more details and explanations **both directly in the rebuttal and in our re-uploaded paper**, to better clarify our notations and proofs. Please refer to our response below for detail.
>
> > *“Authors use a special type of integral, without fully describing what kind of integration they really do. e.g. what they write in eq 4. is in standard mathematical literature known as Line/Contour integral. If that is the case, which parametrization are you using? Please provide more details on the integration part.”*
>
> **A1:**  This is a typo. In our Eq.(1) and Eq.(4), what we want to use is the **standard integral**. Specifically, in Eq.(1) the integral is over each $\omega_t$, $t=1,2,\cdots,T$, and in Eq.(4) the integral is over $\omega_{T+1}$. We have revised all the typos of this kind (by replacing the Line/Contour integral notation with the standard integral one) and added a clear explanation about the integral we use, which can be found in **Eq.(1) and Eq.(4)** in our re-uploaded version.
>
> > *“Can you provide empirical evidence that your Theorem 2 holds?"*
>
> **A2:** According to our Definition 3, the generalization error is defined as the loss for each method on the **future domain** (namely, $\mathcal{D}_{T+1}$). In our experiments, we set our test set as the future domain on all seven datasets in Table 1. Hence, the performance of DRAIN on the test set of each regression/classification dataset in Table 1 **can** well verify our Theorem 2 empirically.
>
> Specifically, “offline" and “IncFinetune" corresponds to the offline and online baseline as defined in Definition 1, respectively. As shown in Table 1, DRAIN always achieves the best performance (excl. ONP), and the online baseline outperforms the offline baseline on four out of six datasets (excl. ONP). Hence, our results in Table 1 can largely support our theory that $err(DRAIN)<err(Online)<err(Offline)$, where $err(\cdot)$ refers to the error on the test set, namely the generalization error.
>
> We will add the explanations above into our future version if given more space after acceptance. Thanks.
>
> > *“Regarding assumption 1 the concept drift is Gaussian. Can you try to quantify this empirically?”*
>
> **A3:** Assumption 1 is over the prior distribution for $q_{\theta}(\cdot)$, where it is assumed that **the initialization for the recurrent structure $q_{\theta}$ follows a Gaussian distribution**, which is widely used in Bayesian statistics ([1] *Bishop, Christopher M., and Nasser M. Nasrabadi. Pattern recognition and machine learning. Vol. 4. No. 4. New York: Springer, 2006*) due to its beautiful properties and convenience to use, and we are following all existing work to consider the Gaussian prior in our analyses.
>
> > *“Eq 16 you add one more assumption on the expected L2 error for your model. Why is this assumption not in the statement of the Theorem?”*
>
> **A4:** The "assume" at the beginning of the sentence before Eq.(16) is misleading. Actually, Eq.(16) can be derived based on the fact that the LSTM unit has been trained to converge on the training domains. In other words, **the converged LSTM unit should have a more accurate prediction compared with the randomly initialized one**. We have revised this part in our Proof 2 to make sure there is no clarity issue now. Please refer to **Proof 2** in the appendix of our re-uploaded version for detail.
>
> > *“Can you clarify part on the mild assumptions for the Lipschitz continuity?”*
>
> **A5:** The Lipschitz assumption we consider is that, given some input $x$ for neural network $g_{\omega}(\cdot)$, we have $L_1  \big\Vert \omega - \omega^{\prime} \big\Vert_2 < \big\Vert g_{\omega}(x) - g_{\omega^{\prime}}(x) \big\Vert < L_2  \big\Vert \omega - \omega^{\prime}  \big\Vert$, where $\omega$ and $\omega^{\prime}$ are two arbitrarily different parameters for $g$. We have revised our Proof 2 to clarify the definition of the Lipschitz continuity. We have also clarified this assumption in our Theorem 2 in the main text. Please refer to **Proof 2** in the appendix of our re-uploaded version for detail.

---

> > ### Comment · Reviewer_vJXs · 2022-12-12
> > **reply to authors**
> >
> > Thank you for the clarifications and corrections. I think that proofs and statements are more clear.
> > Regarding Theorem 2, and empirical proof of it i.e. Table 2 in my view needs more clarifications. E.g. which assumptions of Theorem 2 are/could not be met so that on ONP dataset statement on the generalisation errors are not met. I have changed my rating from 5 to 6 for the recommendation.

---

> > > ### Author Response · Authors · 2022-12-12
> > > **Follow-up Response from Author**
> > >
> > > Dear Reviewer,
> > >
> > > Thanks again for your suggestion and kind consideration. For your question that *“which assumptions of Theorem 2 are/could not be met so that on ONP dataset statement on the generalization errors are not met,”* our best answer is the ONP dataset potentially does not satisfy our Assumption 1 very well. Specifically, our Assumption 1 assumes the prior distribution of the approximator for the concept drift follows a Gaussian distribution, which, however, may not be well satisfied by the concept drift in the ONP dataset. Since we already know there exist weak/no temporal correlations in the ONP dataset, the inductive bias introduced by our Gaussian prior to learning temporal correlations may not help improve the generalization ability of our proposed method in such cases.
> > >
> > > Yes, we will better clarify our discussion about Table 2 and Theorem 2 in our future manuscript. Thank you.

---

### Author Response · Authors · 2022-12-04
**Warm Reminder from Author**

Dear Reviewers,

We sincerely thank ALL the reviewers for your professional and constructive comments given the rather limited reviewing time for this year's ICLR. We have provided detailed comments in rebuttal and revised our paper to better clarify our theoretical analysis. Hope we have addressed all of your concerns.

Just a warm reminder that the main discussion period is coming to an end (**within a week**.) Please let us know if you have any follow-up questions. We will be happy to answer them :)

Best Regards,

Authors

---

### Decision · Program_Chairs · 2023-01-20

**Decision:**

Accept: notable-top-5%

**Justification For Why Not Higher Score:**

N/A

**Justification For Why Not Lower Score:**

Novel method with comprehensive experiments

**Metareview: Summary, Strengths And Weaknesses:**

This paper has proposed a novel Bayesian framework to explicitly model the concept drift over time. Specifically a recurrent neural network is employed to learn the evolving dynamics of model parameter distribution based on how data distribution evolves across the domains. It conducted theoretical analysis and experimental analysis, which demonstrate it performs better than the state-of-the-art methods across multiple classification and regression datasets.


**Note From Pc:**

if the above contains the word "oral" or "spotlight" please see: "oral" presentation means -> notable-top-5% and "spotlight" means -> notable-top-25%. As stated in our emails, we are disassociating presentation type from AC recommendations

**Summary Of Ac-Reviewer Meeting:**

No need to conduct meeting as reviewers generally like this paper